# Mining for Oxysterols in *Cyp7b1^−/−^* Mouse Brain and Plasma: Relevance to Spastic Paraplegia Type 5

**DOI:** 10.3390/biom9040149

**Published:** 2019-04-13

**Authors:** Anna Meljon, Peter J. Crick, Eylan Yutuc, Joyce L. Yau, Jonathan R. Seckl, Spyridon Theofilopoulos, Ernest Arenas, Yuqin Wang, William J. Griffiths

**Affiliations:** 1Swansea University Medical School, ILS1 Building, Singleton Park, Swansea SA2 8PP, UK; a.meljon@qub.ac.uk (A.M.); peter.crick@gmail.com (P.J.C.); eylan.yutuc@swansea.ac.uk (E.Y.); s.theofilopoulos@swansea.ac.uk (S.T.); y.wang@swansea.ac.uk (Y.W.); 2Institute for Global Food Security, Queens University Belfast, Stranmillis Road, Belfast BT9 5AG, UK; 3Endocrinology Unit, BHF Centre for Cardiovascular Science, The Queen’s Medical Research Institute, University of Edinburgh, 47 Little France Crescent, Edinburgh EH16 4TJ, UK; Joyce.Yau@ed.ac.uk (J.L.Y.); J.Seckl@ed.ac.uk (J.R.S.); 4Laboratory of Molecular Neurobiology, Department of Medical Biochemistry and Biophysics, Karolinska Institutet, SE-17177 Stockholm, Sweden; ernest.arenas@ki.se

**Keywords:** cytochrome P450, CYP7B1, hereditary spastic paraplegia, SPG5, 25-hydroxycholesterol, 7α,25-dihydroxycholesterol, cholestenoic acid, cholesterol, liquid chromatography–mass spectrometry, multistage fragmentation

## Abstract

Deficiency in cytochrome P450 (CYP) 7B1, also known as oxysterol 7α-hydroxylase, in humans leads to hereditary spastic paraplegia type 5 (SPG5) and in some cases in infants to liver disease. SPG5 is medically characterized by loss of motor neurons in the corticospinal tract. In an effort to gain a better understanding of the fundamental biochemistry of this disorder, we have extended our previous profiling of the oxysterol content of brain and plasma of *Cyp7b1* knockout (-/-) mice to include, amongst other sterols, 25-hydroxylated cholesterol metabolites. Although brain cholesterol levels do not differ between wild-type (wt) and knockout mice, we find, using a charge-tagging methodology in combination with liquid chromatography–mass spectrometry (LC–MS) and multistage fragmentation (MS^n^), that there is a build-up of the CYP7B1 substrate 25-hydroxycholesterol (25-HC) in *Cyp7b1-/-* mouse brain and plasma. As reported earlier, levels of (25R)26-hydroxycholesterol (26-HC), 3β-hydroxycholest-5-en-(25R)26-oic acid and 24S,25-epoxycholesterol (24S,25-EC) are similarly elevated in brain and plasma. Side-chain oxysterols including 25-HC, 26-HC and 24S,25-EC are known to bind to INSIG (insulin-induced gene) and inhibit the processing of SREBP-2 (sterol regulatory element-binding protein-2) to its active form as a master regulator of cholesterol biosynthesis. We suggest the concentration of cholesterol in brain of the *Cyp7b1-/-* mouse is maintained by balancing reduced metabolism, as a consequence of a loss in CYP7B1, with reduced biosynthesis. The *Cyp7b1-/-* mouse does not show a motor defect; whether the defect in humans is a consequence of less efficient homeostasis of cholesterol in brain has yet to be uncovered.

## 1. Introduction

Cytochrome P450 (CYP) 7B1 (cytochrome P450 family 7 subfamily B member 1) was first identified in 1995 and found to be primarily expressed in brain in rodents [1]. CYP7B1 is an oxysterol- and steroid- 7α-hydroxylase, accepting many oxysterols and cholestenoic acids as substrates as well as steroids including dehydroepiandrosterone (DHEA) [2,3,4]. In humans, deficiency in the enzyme was first revealed in a ten-week-old boy presenting with severe liver disease [5]. In more recent studies, treatment of another infant with this enzyme deficiency with chenodeoxycholic acid has proved successful in resolving liver disease [6]. *CYP7B1* is expressed in human hippocampus and interestingly *CYP7B1* mRNA is significantly reduced in dentate neurons from Alzheimer’s disease subjects [7].

In mice, deletion of *Cyp7b1* results in a mild phenotype despite elevation of tissue and plasma levels of its oxysterol substrates 25-hydroxycholesterol (25-HC) and 26-hydroxycholesterol, presumably the 25R-epimer, (26-HC, also known as 27-hydroxycholesterol, see Appendix A, Appendix A for abbreviations, common and systematic names) [8]. In light of this mouse data, it was surprising when Tsaousidou et al. found in 2008 that sequence alterations in *CYP7B1* were associated with hereditary spastic paraplegia type 5 (SPG5) in humans [9]. Subsequent studies have confirmed patients suffering from SPG5 have a metabolic phenotype characteristic of inactive CYP7B1 [10,11,12,13]. 

In an effort to understand the differences between mouse and human with respect to defective CYP7B1, we embarked on a sterolomic investigation of mouse brain and plasma exploiting enzyme-assisted derivatization for sterol analysis (EADSA) and liquid chromatography–mass spectrometry (LC–MS) with multistage fragmentation (MS^n^). We previously found that in *Cyp7b1* knockout (*Cyp7b1-/-*) mouse brain the concentration of cholesterol was similar to that of the wild-type (wt, *Cyp7b1+/+*), as were the levels of 24S-hydroxycholesterol (24S-HC) [14]. On the other hand, concentrations of (25R)26-hydroxycholesterol (26-HC), 3β-hydroxycholest-5-en-(25R)26-oic acid (3β-HCA) and 24S,25-epoxycholesterol (24S,25-EC) were elevated, presumably being CYP7B1 substrates [12,14]. Now, delving deeper into the sterolome, we reveal that 25-HC and 26-hydroxydesmosterol (26-HD) are also elevated in *Cyp7b1-/-* mouse brain, whereas 7α,25-dihydroxysterols are reduced in abundance, but other oxysterols 24R-hydroxycholesterol (24R-HC), 7α- and 7β-hydroxycholesterol (7α-HC, 7β-HC) do not change in concentration between the two genotypes. We suggest that elevated levels of 25-HC, 26-HC and 24S,25-EC in brain reduce the expression of cholesterol biosynthetic genes by inhibiting the processing of SREBP-2 (sterol regulatory element-binding protein 2) to its active form as a master transcription for the mevalonate pathway [15], thereby reducing cholesterol synthesis and compensating for its reduced metabolism (via the CYP7B1 pathway), thus maintaining cholesterol levels in *Cyp7b1-/-* mouse brain at wild-type levels. 25-HC, 24S,25-HC, 3β-HCA, and in some studies 26-HC, have all been found to be ligands to the liver X receptors (LXRα, NR1H3; LXRβ, NR1H2) [16,17,18], activation of which increases the expression of ATP-binding cassette transporter A1 (ABCA1) and apolipoprotein E (APOE), transporter and carrier proteins important for maintaining correct sterol levels in neurons [19] and avoiding overload by potentially toxic oxysterols.

## 2. Materials and Methods 

### 2.1. Animals

Brain and plasma samples were from male mice of 13 and 23 months of age. *Cyp7b1-/-* and *Cyp7b1+/+* mice were littermates generated from *Cyp7b1+/-* crosses at the University of Edinburgh animal facilities [3]. All mice were housed under standard conditions (7:00 am to 7:00 pm light/dark cycle, 21 °C) with food and water available ad libitum. Tissue sampling was performed under the aegis of the UK Scientific Procedures (Animals) Act, 1986, amended in 2012 to comply with the European Directive 2010/63/EU. The study was conducted under PPL No.70/7870 with prior approval from the University of Edinburgh Animal Welfare and Ethical Review Body. All mice were sacrificed in the morning by cervical dislocation, trunk blood collected and brains removed, frozen on powdered dry ice and stored at −80 °C.

### 2.2. Sample Preparation

Sterols including oxysterols were extracted from brain as described in [12,14,20,21]. In brief, mouse brain was homogenised in ethanol containing isotope-labelled internal standards and oxysterol- and cholesterol-rich fractions separated by solid phase extraction (SPE) on a reversed phase C_18_ column. The oxysterol-rich fraction was then divided into two aliquots (a and b) and the first treated with cholesterol oxidase enzyme from *Streptomyces* sp. to oxidise 3β-hydroxy-5-ene groups to 3-oxo-4-ene equivalents, suitable for subsequent derivatisation with the Girard P (GP) reagent (i.e., fraction a). The second aliquot of the oxysterol fraction was treated with GP reagent directly, in the absence of cholesterol oxidase (i.e., fraction b). This allowed the differentiation of oxysterols with a native 3-oxo-4-ene structure (fraction b) from those with a 3β-hydroxy-5-ene structure (i.e., (fraction a)–(fraction b)). The cholesterol-rich fraction was treated separately, but in the same way as the oxysterol-rich fraction.

Plasma samples were prepared as described in Autio et al. and Crick et al. by extraction into ethanol followed by SPE to separate cholesterol- and oxysterol-rich fractions [22,23]. The oxysterols were then derivatised with GP reagent with (fraction a), or without (fraction b), prior oxidation by cholesterol oxidase. For plasma analysis, two GP reagents were used [^2^H_0_]GP and [^2^H_5_]GP with either fraction a or with fraction b, respectively. This allowed the duplex LC-MS analysis of oxysterol fractions prepared with (fraction a) or without (fraction b) treatment with cholesterol oxidase (see Appendix A, Appendix A) [24].

### 2.3. Analysis

The oxidised/derivatised oxysterol-rich fractions were analysed by LC-MS (MS^n^) using an Ultimate 3000 LC system (Thermo Fisher Scientific, Loughborough, UK) and LTQ-Orbitrap mass spectrometer (Thermo Fisher Scientific, Loughborough, UK) as described in Meljon et al. [20] and Crick et al. [23,24]. In brief, GP-derivatised oxysterols were separated on a reversed phase Hypersil Gold C_18_ column (Thermo Fisher Scientific) using a methanol/acetonitrile/0.1% formic acid gradient. The eluent was directed to an electrospray ionisation source (ESI) and analysed by high-resolution (60,000 at *m/z* 400) MS and MS^3^ ([M]^+^→[M-Py]^+^→, where “-Py” corresponds to the loss of the pyridine group from the molecular ion M^+^) scans performed in parallel in the Orbitrap and LTQ linear ion-trap, respectively. Quantification was performed using the isotope dilution method.

## 3. Results

### 3.1. Brain

#### 3.1.1. Oxysterols

We previously reported the levels of 24S-HC, 24S,25-EC [14], 26-HC, 3β-HCA and the combination of 7α,26-dihydroxycholesterol (7α,26-diHC) and 7α,26-dihydroxycholest-4-en-3-one (7α,26-diHCO) (i.e., fraction a, [12]) in *Cyp7b1-/-* and *Cyp7b1+/+* mouse brain but, at that time, did not analyse 25-hydroxylated sterols or other 7α-hydroxylated sterols, or oxysterols derived from cholesterol precursors. Here we report new data for these additional sterols. The complete data set is presented in Figure 1 and Appendix A, Appendix A.

In the *Cyp7b1+/+* mice, the concentration of 25-HC is below the limit of detection of the LC-MS method (<0.01 ng/mg); however, in the *Cyp7b1-/-* mice, the level of 25-HC is significantly elevated (*p* < 0.001) in both 13 (0.89 ± 0.14 ng/mg, mean ± SD) and 23 (2.09 ± 0.45 ng/mg) month-old mice. This is particularly evident when using an extended chromatographic gradient (33 min) which resolves 25-HC from other side-chain hydroxycholesterols (Figure 2). Unlike 24S-HC, 24R-HC is only a minor oxysterol in mouse brain [21], but like 24S-HC its concentration in the *Cyp7b1-/-* mouse brain was not found to differ from that measured in wild-type (wt) animals (0.20 ± 0.09 ng/mg and 0.30 ± 0.03 ng/mg at 13 and 23 months, respectively, in wt).

As reported earlier [12], the concentration of 26-HC is elevated in the *Cyp7b1-/-* mice (Figure 2a,b). This oxysterol is formed by oxidation of cholesterol by CYP27A1 (cytochrome P450 family 27 subfamily A member 1) [25]. CYP27A1 can also oxidize desmosterol (24-DHC, 24-dehydrocholesterol) to 26-HD [26] and its cis (*Z*) and trans (*E*) geometric isomers have been reported to be present in newborn mouse brain [20]. Although only minor oxysterols, both isomers are found to be elevated in concentration in *Cyp7b1-/-* mouse brain, most prominently in 23-month-old animals, where the concentration of the (*E*) isomer was raised from 0.01± 0.00 ng/mg in the wt to 0.04 ± 0.00 ng/mg in the *Cyp7b1-/-* animals (*p* < 0.001, Figure 1 and Figure 3).

7α-HC is also a minor oxysterol in wt mouse brain (0.01 ng/mg), but in brain from the *Cyp7b1-/-* 23-month-old mice its concentration is raised to 0.02 ± 0.00 ng/mg (*p* < 0.05). Likewise, 7β-HC, another minor oxysterol, is more abundant in brain from the 23-month-old *Cyp7b1-/-* mice (0.03 ± 0.01 ng/mg) than the wt animals of the same age (0.02 ± 0.00 ng/mg, *p* < 0.05).

7α,25-Dihydroxycholesterol (7α,25-diHC) and 7α,25-dihydroxycholest-4-en-3-one (7α,25-diHCO) are present at low levels in brain (Figure 1). Using the EADSA methodology, these two dihydroxysterols can be measured in combination (fraction a) more accurately than for the 3β-hydroxy or 3-oxo analytes alone (fraction a–fraction b and fraction b, respectively). This is also true for 7α,26-diHC and 7α,26-diHCO [21]. 7α,25-diHC/7α,25-diHCO, like many other GP-derivatised oxysterols elute as *syn* and *anti* conformers giving a pair of peaks. Unfortunately, the first of these peaks co-elutes with the first peak from the 7α,24-dihydroxysterols, 7α,24-dihydroxycholesterol (7α,24-diHC)/ 7α,24-dihydroxycholest-4-en-3-one (7α,24-diHCO), although the second peaks from 7α,25- and 7α,24-dihydroxysterols are resolved (Figure 4). However, as the first peaks are dominant, we decided to measure concentrations of 7α25- and 7α,24-sterols in combination. In *Cyp7b1-/-* mouse brain, we found that at both 13 and 23 months of age the concentration of the 7α,24/25-dihydroxysterols is reduced from 0.01± 0.00 ng/mg and 0.02 ± 0.00 ng/mg in the wt to 0.00 ± 0.00 ng/mg and 0.01 ± 0.00 ng/mg, in the *Cyp7b1-/-* animals. Surprisingly, in our earlier study, we did not find that the 7α,26-dihydroxysterol concentrations varied with genotype [12].

The biochemical results presented above reveal an increased concentration of oxysterol ligands to the LXRs and to INSIG (insulin-induced gene) in brain of the *Cyp7b1-/-* mouse. The biological consequence of this is likely to be increased transport of cholesterol between cells through enhanced expression of ABC transporters and APOE [27], and of reduced cholesterol biosynthesis via inhibition of the SREBP-2 pathway [15].

#### 3.1.2. Sterols

We have previously reported that the concentration of cholesterol in brain does not vary between the *Cyp7b1+/+* and *Cyp7b1-/-* genotypes (10.60 ± 1.43 ng/mg at 13 months old and 15.81 ± 0.65 at 23 months old in wt animals) [14]. Measurements of cholesterol precursors is limited in the current LC-MS setting by the dynamic range of the chromatographic system, where injection of sufficient sample to achieve peak areas of the necessary size to allow accurate measurements of cholesterol precursors overloads the LC column with cholesterol. As the goal of the current study was primarily to measure oxysterols, we only measured cholesterol precursors in a single brain of each genotype from 13-month-old animals. Interestingly, the levels of both desmosterol and 8(9)-dehydrocholesterol (8-DHC), an enzymatically derived isomer of 7-dehydrocholesterol (7-DHC), are reduced in the *Cyp7b1/-/-* mouse (Figure 5).

Although cholesterol levels do not differ between the two genotypes, the reduction of cholesterol precursors in *Cyp7b1-/-* mouse brain does suggest a reduced cholesterol synthesis as a result of oxysterol-induced inhibition of the SREBP-2 pathway. 

### 3.2. Plasma

We have previously reported the concentrations of 26-HC and its downstream metabolites, 7α,26-diHC/7α,26-diHCO, 3β-HCA and 7αH,3O-CA in plasma [12]. Here, we extend that study by reporting concentrations of further oxysterols, including 24S,25-EC, 25-HC and other 7-oxidised metabolites (Figure 6 and Appendix A, Appendix A).

As in brain, the concentration of total 24S,25-EC is significantly higher (*p* < 0.001) in plasma of the *Cyp7b1-/-* mice (10.82 ± 2.74 ng/mL at 13 months, 8.63 ± 1.11 ng/mL at 23 months) than *Cyp7b1+/+* mice (1.16 ± 0.98 ng/mL at 13 months, 1.52 ± 0.59 ng/mL at 23 months, Appendix A, Appendix A). This is also true for 25-HC where the plasma concentration in the *Cyp7b1-/-* mice (36.52 ± 7.04 ng/mL at 13 months, 42.19 ± 20.21 ng/mL at 23 months) is also higher (*p* < 0.001 at 13 months, *p* < 0.01 at 23 months) than in the *Cyp7b1+/+* mice (0.74 ± 0.24 ng/mL at 13 months, 0.85 ± 0.26 ng/mL, Appendix A, Appendix A). Unlike the situation for 24S,25-EC, 25-HC and 26-HC [12], where plasma concentrations are elevated in *Cyp7b1-/-* mice, the concentration of 24S-HC does not vary between genotypes (in wt animals 2.02 ± 0.54 ng/mL at 13 months and 1.78 ± 0.58 ng/mL at 23 months). This is also true for the minor oxysterol 22R-hydroxycholesterol (22R-HC, in wt animals 0.17 ± 0.11 ng/mL at 13 months and 0.31 ± 0.21 ng/mL at 23 months).

CYP7A1 (cytochrome P450 family 7 subfamily A member 1) is the enzyme responsible for the formation of 7α-HC from cholesterol [28]. The concentrations of 7α-HC and its metabolite 7α-hydroxycholest-4-en-3-one (7α-HCO) show considerable variation within each genotype at both ages and no significant differences between genotypes are discernible. This is also true for 7-oxocholesterol (7-OC), which may be formed from 7-DHC by CYP7A1 [29] and for 7β-hydroxycholesterol (7β-HC) which can be formed from 7-OC by the enzyme HSD11B1 (hydroxysteroid 11β-dehydrogenase 1) [30]. An added complication is that each of these oxysterols may be formed from cholesterol via free radical oxidation, perhaps during sample preparation or storage [31]. 

In plasma it was possible to differentiate the concentration of 7α,25-diHCO from that of 7α,25-diHC which is essentially absent in mice of both genotypes. In the 13-month-old mice, 7α,25-diHCO was at or below the detection limit in plasma from the *Cyp7b1-/-* animals but present in *Cyp7b1+/+* plasma (0.44 ± 0.30 ng/mL, *p* = 0.01, Appendix A, Appendix A). 7α,26-diHCO was observed in plasma from both genotypes at both ages, and at 13 months the concentration difference between *Cyp7b1+/+* (1.10 ± 0.44 ng/mL) and *Cyp7b1-/-* (0.64 ± 0.14 ng/mL) was almost significant *p* = 0.055). At 13 months, the concentration of the downstream metabolite 7αH,3O-CA was almost significantly different (*p* = 0.051, Appendix A, Appendix A) between the wt (25.89 ± 10.91 ng/mL) and the *Cyp7b1-/-* mice (14.21 ± 3.76 ng/mL) [12].

7α,12α-Dihydroxycholesterol (7α,12α-diHC) and 7α,12α-dihydroxycholest-4-en-3-one (7α,12α-diHCO) are observed in plasma of both genotypes. There is also evidence for the presence of 12α-hydroxycholesterol (12α-HC).

7αH,3O-CA can be metabolized further to the CoA thioesters of 7α,24R-dihydroxy-3-oxocholest-4-en-(25R)26-oic and 7α-hydroxy-3,24-*bis*oxocholest-4-en(25R)26-oic acid in the peroxisome [32,33]. The latter compound is unstable in our analytical system and decomposes by loss of C-26 to give 7α-hydroxy-26-*nor*cholest-4-en-3,24-dione (7αH,26-nor-C-3,24-diO) which we observe in mouse plasma. An alternative route of metabolism of 7αH,3O-CA is 12α-hydroxylation in the endoplasmic reticulum by CYP8B1 (cytochrome P450 family 8 subfamily B member 1) to give 7α,12α-dihydroxy-3-oxocholest-4-en-(25R)26-oic acid (7α,12α-diH,3O-CA) which we also observe in plasma. Unfortunately, we do not have statistical data for 7αH,26-nor-C-3,24-diO or 7α,12α-diH,3O-CA as only one (or two) animals were analysed for each genotype at each age in this study. This is also true for 12α-HC, 7α,12α-diHC and 7α,12α-diHCO.

Besides brain, *Cyp7b1* is expressed in liver, lung, kidney and reproductive tract in mice [34]. Hence, the origins of the quantitative differences observed in plasma between *Cyp7b1-/-* and *Cyp7b1+/+* mice are likely to be multifactorial. Nevertheless, the enhanced abundance of 25-HC, 26-HC, 24S,25-EC and 3β-HCA in *Cyp7b1-/-* mouse plasma confirms these molecules as substrates for CYP7B1. A decreased abundance in 7α,26-diHCO and 7αH,3O-CA might also have been expected in the *Cyp7b1-/-* mouse; however, these two metabolites can alternatively be synthesized via a pathway initiated by CYP7A1 which is elevated in the *Cyp7b1-/-* mouse [8].

## 4. Discussion

CYP7B1 is a known 7α-hydroxylase towards 25-HC, 26-HC and 3β-HCA [8,35]. CYP7B1 is expressed in brain [1,2,3] and data presented here demonstrates its activity in mouse brain towards 25-HC, as seen by elevated levels of 25-HC and reduction in levels of its metabolite 7α,25-diHCO in brain of the *Cyp7b1-/-* mouse. 26-HD exists as (*Z*) and (*E*) isomers and both are present at very low levels in brain [20]; however, their elevated abundance in the *Cyp7b1-/-* mouse indicates that, like 26-HC, 26-HD(*Z*) and 26-HD(*E*) are substrates for CYP7B1 in brain.

The major route for cholesterol removal from brain is via oxidation by CYP46A1 to 24S-HC [36], although other mechanisms also exist, most likely involving CYP27A1 and CYP7B1 [37,38]. CYP7B1 has little activity towards 24S-HC, the product of CYP46A1 oxidation of cholesterol, where 7α-hydroxylation is catalysed by CYP39A1 (cytochrome P450 family 39 subfamily A member 1) [39]. By blocking one of the pathways of cholesterol metabolism in brain by deletion of *Cyp7b1,* it may be predicted that either cholesterol or 24S-HC levels would increase; however, neither of these events occurs. This can be explained by down-regulation of cholesterol biosynthesis by inhibition of the SREBP-2 pathway. Besides 25-HC, 26-HC and 24S,25-EC are elevated in brain of the *Cyp7b1-/-* mouse. Each will interact with the endoplasmic reticulum resident protein INSIG and block transport by SCAP (SREBP cleavage-activating protein) of SREBP-2 to the Golgi for processing to its active form as the master transcription factor for genes of the cholesterol biosynthesis pathway [15]. Although we do not prove this explanation here, there is no reason to expect that the INSIG–SCAP–SREBP regulatory mechanism differs in cells expressing or not expressing CYP7B1, other than with respect to the abundance of oxysterols and steroids. 25-HC, 26-HC, 24S,25-EC and 3β-HCA are also ligands to the LXRs [16,17,18], both of which are expressed in brain [40]. LXR target genes include *Abca1*, which codes for the cholesterol efflux pump ABCA1, and *Apoe* which codes for the apolipoprotein APOE, which mediates transport of cholesterol within the brain [19]. The lack of a major neurological phenotype in the *Cyp7b1-/-* mouse, despite the absence of a key sterol metabolizing enzyme, is likely a consequence of the safety net provided by side-chain oxysterols, INSIG–SCAP–SREBP and LXR working in concert to avoid detrimental concentrations of cholesterol and of oxysterols building up in neuronal cells.

In humans, deficiency in CYP7B1 leads to SPG5, a form of spastic paraplegia, defined by progressive neurodegeneration of corticospinal tract motor neurons. One hypothesis to explain the different phenotype between CYP7B1-deficient mouse and human is that, in humans, oxysterol levels in the central nervous system are less well regulated and reach a level toxic to corticospinal tract neurons resulting in SPG5 [11]. In fact, Schöls et al showed that the concentration of total (the sum of esterified and non-esterified) 26-HC in serum is correlated with SPG5 disease severity and duration [11]. They found that at concentrations of 25-HC and 26-HC close to those found in serum of SPG5 patients, both oxysterols were toxic towards motor neuron-like cells as well as cortical neurons derived from human induced pluripotent stem cells (iPSCs) [11]. It should be noted that concentrations of these molecules in serum from SPG5 patients (total 25-HC 177.9 ± 77.0 ng/mL, mean ± SD, *n* = 19; 26-HC 878.2 ± 207.1 ng/mL, *n* = 19) were far higher than the value measured for 26-HC in cerebrospinal fluid (CSF, 12.9 ± 4.3 ng/mL, *n* = 19, concentration of 25-HC in CSF were not reported) and that the free molecules (i.e., non-esterified oxysterols), which were used in toxicity tests, are likely to be present at concentrations of about 10% of the sum of non-esterified and esterified molecules. 3β-HCA was also found to be toxic to the cells but at concentrations higher than those found in SPG5 serum [11].

The concentrations of 25-HC, 26-HC and 3β-HCA (363.1 ± 82.1 ng/mL, *n* = 19) reported by Schöls et al. in SPG5 serum [11] are considerably higher than those measured here in plasma for the *Cyp7b1 -/-* mouse (25-HC, 36.52–42.19 ng/mL; 26-HC, 23.80–16.93 ng/mL; 3β-HCA, 7.32–6.69 ng/mL, numbers are means at the two ages). However, we measured the concentrations of the free non-esterified molecules, whereas Schöls et al. measured the total non-esterified plus esterified molecules [11]. A better comparison is provided by the data reported by Theofilopoulos et al., where the values for the non-esterified molecules were measured as 25-HC 49.40 ± 11.38 ng/mL, 26-HC 97.75 ± 7.28 ng/mL and 368.40 ± 65.27 ng/mL, *n* = 9, in SPG5 plasma/serum [12]. The concentrations of non-esterified 26-HC and 3β-HCA in human plasma/serum are considerably greater than those in the *Cyp7b1-/-* mouse, and this may explain the presence of a motor neuron disease phenotype in SPG5 patients but not in the *Cyp7b1-/-* mouse.

Theofilopoulos et al. also measured the concentration of 3β,7α-diHCA in human CSF and plasma [12]. They found that in SPG5 patients the levels of this acid were significantly reduced in both fluids compared to controls and, based on its neuroprotective effects towards oculomotor neurons, suggested that its reduced levels in SPG5 patients may contribute towards their motor neuron loss [12]. In the current study, we failed to detect 3β,7α-diHCA in either mouse brain or plasma. This is probably a consequence of its greater rate of metabolism to 7αH,3O-CA by HSD3B7 (3β-hydroxysteroid dehydrogenase type 7) in mice than in humans.

CYP7B1 is an essential enzyme in the acidic pathway of bile acid biosynthesis [28,41]. Surprisingly, this is not reflected in the plasma concentration of 7αH,3O-CA in mice where we did not observe a statistical difference between the two genotypes. This can be explained by the existence of an alternative route to 7αH,3O-CA avoiding CYP7B1 and rather utilizing 7α-hydroxylation of cholesterol by CYP7A1 [33,41]. The resulting 7α-HC is then oxidized at C-3, before or after oxidation at C-26 by CYP27A1 to give 7αH,3O-CA.

As mentioned in Section 3.1.2, the level of cholesterol was measured in each of the mouse brains from both genotypes. Its precursor desmosterol and the 7-DHC isomer 8-DHC are also present in brain but at much lower levels (see Appendix A, Appendix A). The limited dynamic range of the LC-MS system makes the simultaneous analysis of cholesterol and of its precursors challenging. As the main focus in this study was on oxysterols and not cholesterol precursors, these sterols were not investigated thoroughly. Nevertheless, the data from single animals suggest reduced levels of cholesterol precursors in brain of the *Cyp7b1-/-* animals. Further studies are required to confirm or refute this suggestion. Similarly, further studies are required to investigate the concentration in plasma of 12α-hydroxy metabolites, which in this work were detected but not measured in all animals. Their presence or absence in brain also requires further investigation.

## 5. Conclusions

Despite the elimination of a metabolic pathway for cholesterol removal in brain of the *Cyp7b1-/-* mouse, the concentration of cholesterol is not elevated. This can be explained by a reduction in its synthesis due to a down-regulation of the cholesterol biosynthesis pathway by inhibition of SREBP-2 processing by side-chain oxysterols elevated in concentration in brain of the *Cyp7b1-/-* mouse. Unlike human CYP7B1-deficiency, the *Cyp7b1-/-* mouse does not have a pronounced neurological phenotype. Our results support the view that in mice the production of neurotoxic sterol metabolites is less marked than in humans, with the consequence that mice do not present with spastic paraplegia, while humans do.

## 6. Patents

The derivatization method described in this manuscript is patented by Swansea University, Swansea UK (US9851368B2) and licensed by Swansea Innovations to Avanti Polar Lipids and to Cayman Chemical Company.

## Figures and Tables

**Figure 1 biomolecules-09-00149-f001:**
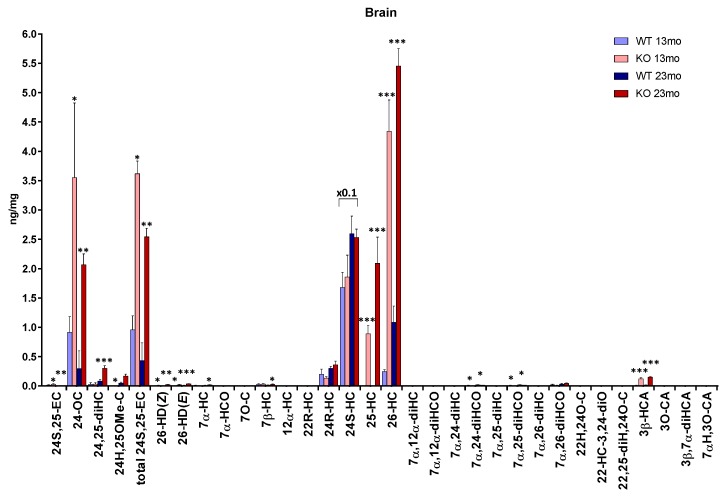
Concentrations of oxysterols in mouse brain. To maintain data to a single y-axis, a magnification factor of 0.1 has been applied to 24S-HC. Data for 24S-HC and total 24S,25-EC is from [14] and for 26-HC, 3β-HCA, 7α,26-diHC + 7α,26-diHCO, 3β,7α-dihydroxycholest-5-en-(25R)26-oic acid (3β,7α-diHCA) and 7α-hydroxy-3-oxocholest-4-en-(25R)26-oic acid (7αH,3O-CA) is from [12]. At 13 months of age, *n* = 3 animals, and at 23 months of age, *n* = 4 animals, for each genotype. Student’s *t*-tests were performed between wild-type (wt) and *Cyp7b1-/-* animals of the same age and significance indicated by *, *p* < 0.05; **, *p* < 0.01; ***, *p* < 0.001 placed above the appropriate bars for the transgenic animals. Error bars represent standard deviation (SD). 7α,26-diHC was measured in combination with 7α,26-diHCO. The combined value is applied to the 7α,26-diHCO label only. Similarly, 7α,25-diHC was measured in combination with 7α,25-diHCO. The combined value is applied to the 7α,25-diHCO label only. The 7α,25-dihydroxysterols were incompletely resolved from 7α,24-dihydroxysterols, thus the value given includes values from both dihyroxysterols. 24S,25-EC becomes hydrolysed to 24,25-dihydroxycholesterol (24,25-diHC), undergoes methanolysis to 24-hydroxy,25-methoxycholesterol (24H,25OMe-C) and isomerisation to 24-oxocholesterol (24-OC) during sample preparation. The total 24S,25-EC value is the sum of remaining 24S,25-EC and these three secondary forms.

**Figure 2 biomolecules-09-00149-f002:**
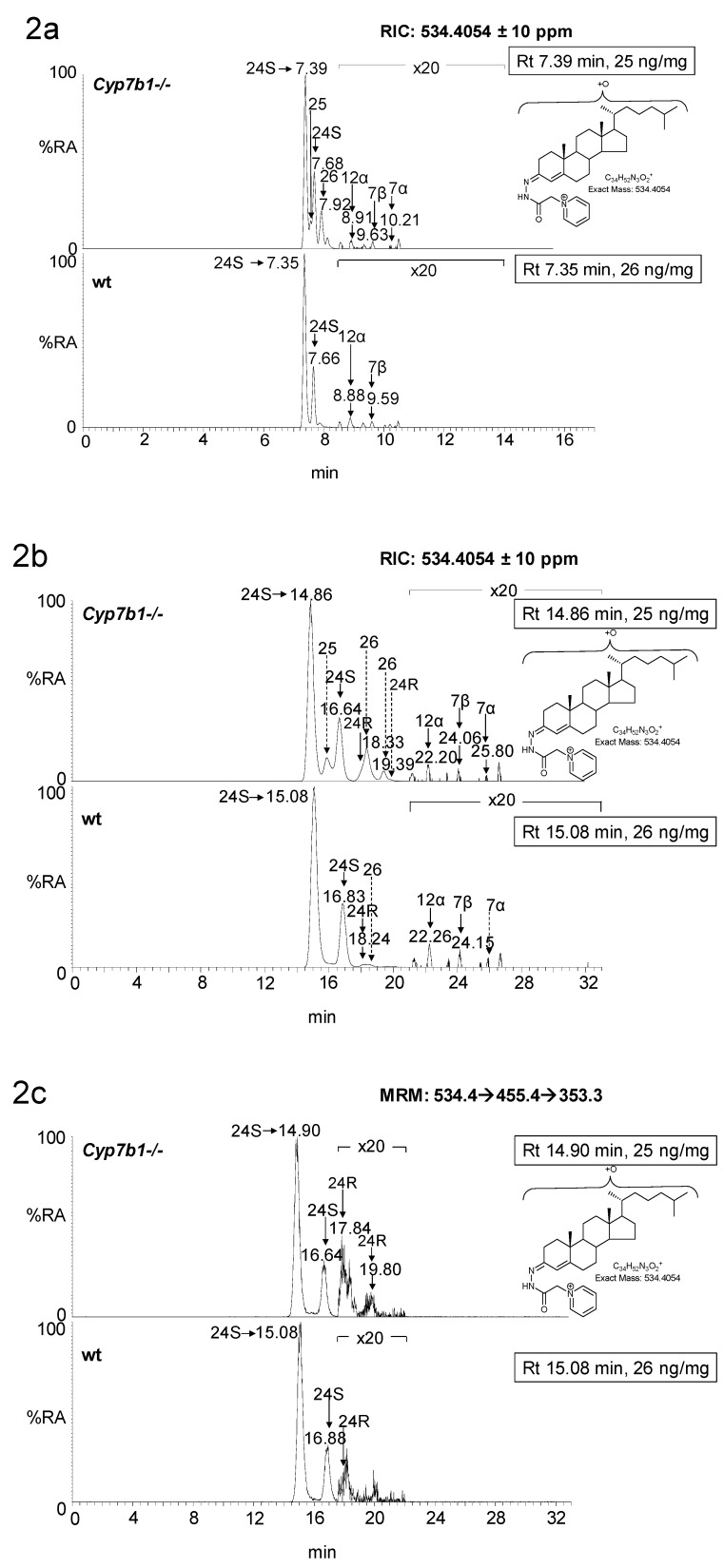
LC-MS(MS^n^) analysis of monohydroxycholesterols in mouse brain. Reconstructed ion-chromatograms (RICs) from analysis of *Cyp7b1-/-* (upper panel) and *Cyp7b1+/+* (wt, lower panel) 23-month-old animals generated from (**a**) 17 min and (**b**) 33 min chromatographic gradients. (**c**) Multiple reaction monitoring (MRM) chromatogram (*m/z* 534.4→455.4→353.3, see Appendix A, Appendix A) from *Cyp7b1-/-* (upper panel) and *Cyp7b1+/+* (lower panel) animals generated from the 33 min chromatographic gradient. Magnification factors in chromatograms (**a**–**c**) are as indicated. Concentration of the analyte indicated by retention time is given in the right-hand corner of the chromatograms. MS^3^ spectra ([M]^+^→[M-Py]^+^→) of (**d**) 24S-HC from a *Cyp7b1+/+* mouse and of (**e**) 24R-HC, (**f**) 25-HC, (**g**) 26-HC and (**h**) 7β-HC and 7α-HC, from a *Cyp7b1-/-* mouse. Many GP-derivatised sterols give *syn* and *anti* conformers which may or may not be chromatographically resolved. Generic fragmentation schemes are illustrated in Appendix A, Appendix A.

**Figure 3 biomolecules-09-00149-f003:**
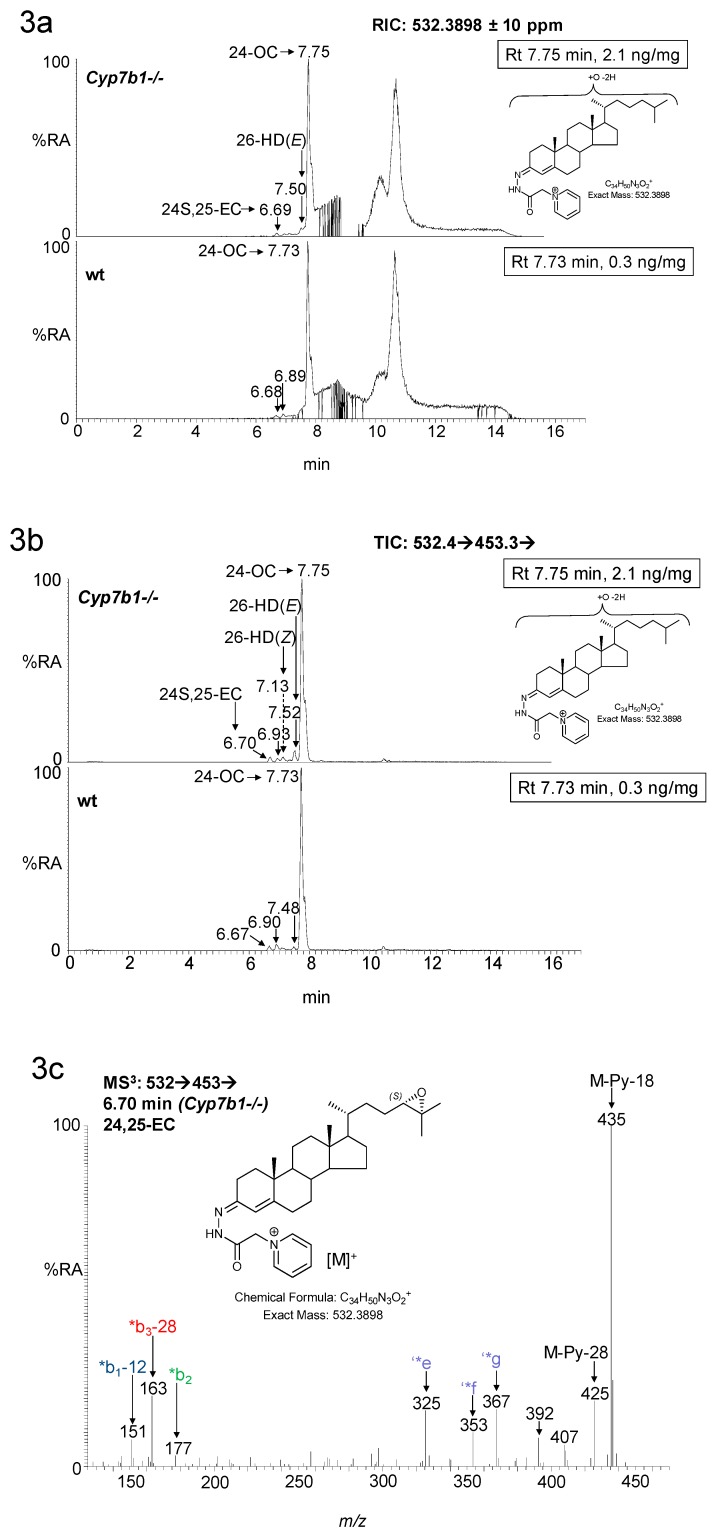
LC-MS(MS^n^) analysis of monohydroxydesmosterols and their isomers 24S,25-EC and 24-OC in mouse brain. (**a**) RICs from analysis of *Cyp7b1-/-* (upper panel) and *Cyp7b1+/+* (wt, lower panel) 23-month-old animals generated from the 17 min gradient. (**b**) Total ion chromatograms (TICs) for the MS^3^ transition [M]^+^→[M-Py]^+^→ using the same gradient as in (**a**). Concentration of the analyte indicated by retention time is given in the right-hand corner of the chromatograms. MS^3^ spectra ([M]^+^→[M-Py]^+^→) of (**c**) 24S,25-EC, (**d**) 26-HD(*Z*), (**e**) 26-HD(*E*) and (**h**) 24-OC from the *Cyp7b1-/-* mouse; and of authentic and of standards of (**f**) 26-HD(*Z*) and 26-HD(*E*) with their (**g**) chromatograms. 24-OC is derived from 24S,25-EC during the sample preparation process. Many GP-derivatised sterols give *syn* and *anti* conformers which may or may not be chromatographically resolved.

**Figure 4 biomolecules-09-00149-f004:**
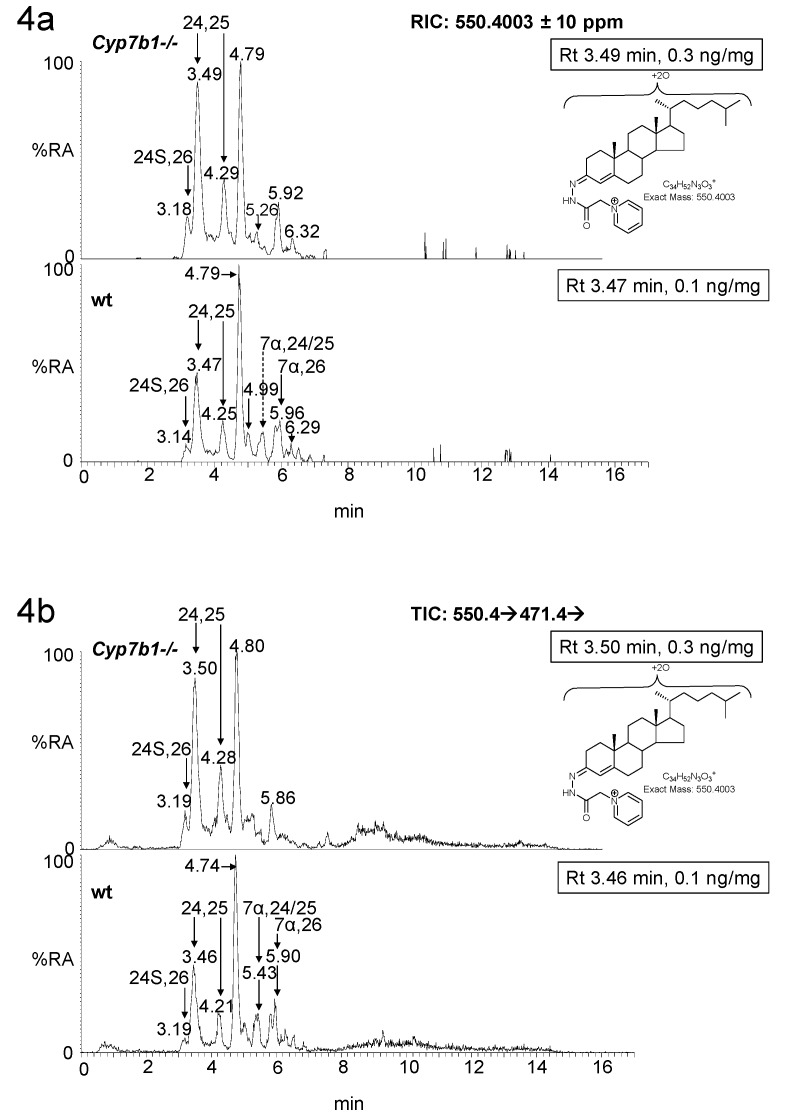
LC-MS(MS^n^) analysis of dihydroxysterols in mouse brain. (**a**) RICs from analysis of *Cyp7b1-/-* (upper panel) and *Cyp7b1+/+* (wt, lower panel) 23-month-old animals, generated from the 17 min gradient. (**b**) TICs for the MS^3^ transitions [M]^+^→[M-Py]^+^→ using the same gradient as in (**a**). Concentration of the analyte indicated by retention time is given in the right-hand corner of the chromatograms. MS^3^ spectra ([M]^+^→[M-Py]^+^→) of (**c**) 24,25-diHC *Cyp7b1-/-* mouse; and (**d**) 7α,24-diHC/7α,24-diHCO, (**e**) 7α,25-diHC/7α,25-diHCO, (**f**) unresolved mixture of 7α,24- and 7α,25-dihydroxysterols, and (**g**) 7α,26-diHC/7α,26-diHCO from the *Cyp7b1+/+* mouse and (**h**) 24S,26-diHC from the *Cyp7b1-/-* mouse. 24,25-diHC is the hydrolysis product of 24S,25-EC. 24S,26-diHC was detected but not quantified. Many GP-derivatised sterols give *syn* and *anti* conformers which may or may not be chromatographically resolved.

**Figure 5 biomolecules-09-00149-f005:**
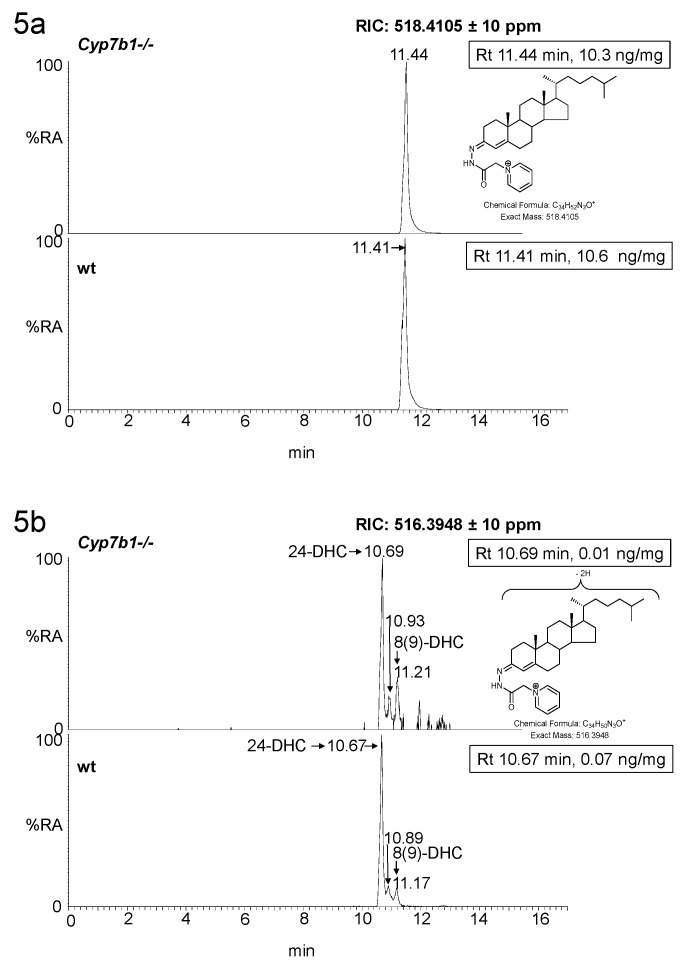
LC-MS(MS^n^) analysis of cholesterol and its precursors in mouse brain. RICs from analysis of *Cyp7b1-/-* (upper panel) and *Cyp7b1+/+* (wt, lower panel) 13-month-old animals, generated from the 17 min gradient. (**a**) RICs for cholesterol. (**b**) RICs for desmosterol (24-DHC) and 8-DHC. Concentration of the analyte indicated by retention time is given in the right-hand corner of the chromatograms. MS^3^ spectra ([M]^+^→[M-Py]^+^→) of (**c**) cholesterol, (**d**) desmosterol and (**e**) 8-DHC from the *Cyp7b1+/+* mouse.

**Figure 6 biomolecules-09-00149-f006:**
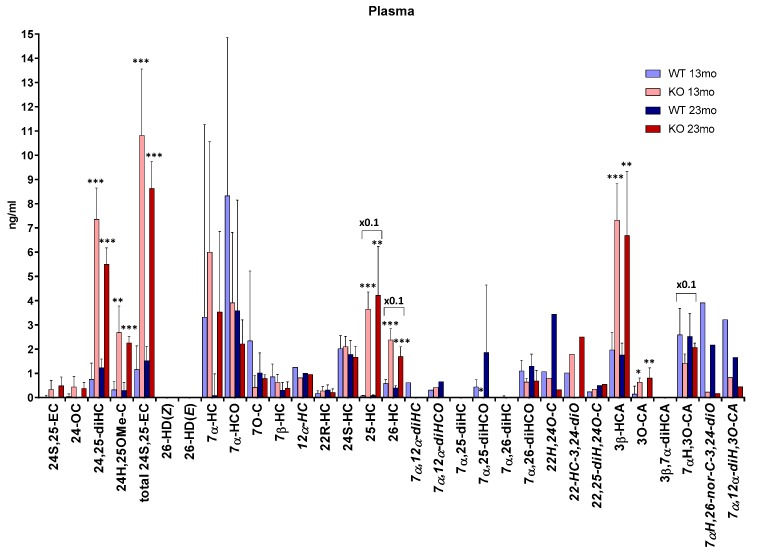
Concentrations of oxysterols in mouse plasma. To maintain data to a single y-axis, a magnification factor of 0.1 has been applied to 25-HC, 26-HC and 7αH,3O-CA. Data for 26-HC, 3β-HCA, 7α,26-diHCO, 3β,7α-diHCA and 7αH,3O-CA is from [12]. At 13 months of age, *n* = 5 animals, and at 23 months of age, *n* = 4 animals, for each genotype (for metabolites indicated by abbreviations in italics, data was obtained from only one or two animals). Student’s *t*-tests were performed between *Cyp7b1+/+* (wt) and *Cyp7b1-/-* animals of the same age and significance indicated by *, *p* < 0.05; **, *p* < 0.01; ***, *p* < 0.001 placed above the appropriate bars for the transgenic animals. Error bars represent standard deviation (SD). 24S,25-EC becomes hydrolysed to 24,25-diHC, undergoes methanolysis to 24H,25OMe-C and isomerisation to 24-OC during sample preparation. The total 24S,25-EC value is the sum of remaining 24S,25-EC and these three secondary forms.

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
