# Peer review of "Mining for Oxysterols in Cyp7b1−/− Mouse Brain and Plasma: Relevance to Spastic Paraplegia Type 5"

_biomolecules, 2019, doi:10.3390/biom9040149_

Round 1

Reviewer 1 Report

The paper entitled “Mining for Oxysterols in Cyp7b1-/- Mouse Brain and  Plasma” by Anna Meljon and coll. suggests that the concentration of cholesterol in brain  of Cyp7b1-/- mouse is maintained by balancing reduced metabolism, as a consequence of a loss in  CYP7B1 with reduced biosynthesis.

The paper is well written and well carried out.

The results are very interesting and suggest to biomedical research groups to uncover  whether the detrimental effects of Cyp7b1 deficiency in human beings is a consequence of a less efficient homeostasis of cholesterol with respect to rodent one.

Author Response

Reviewer 1

We thank this reviewer for his/her generous comments.

Reviewer 2 Report

In the current study, authors have done a great job of characterizing the oxysterol profiles of Cyp7b1 knock-out mouse model by using LCMS in continuation of their previous work. Though experimental design and sample analyses were carefully carried out, there hasn’t been conclusive on whether Cyp7b1 knock-out mouse is a suitable model to study human SPG disease.

Major Concerns, 

1.    Regarding to the title, instead of describing the approach, please give a conclusive statement on the current study, that readers can have the take home message easily.

2.    In the result sections, besides describing the data analysis, please provide biological implication in each section.

3.    Both in the Abstract and Discussion, authors mentioned side-chain oxysterols binds INSIG and inhibit SREBP-2 pathways, please provide direct evidence to show it is the case in the Cyp7b1 knock-out mouse model as well. 

Author Response

We thank the reviewer for his/her positive comments.

1. We have changed the title to reflect the importance of our data to the human disease Hereditary Spastic Paraplegia Type 5.

2. At the end of each result sub-section we have added a summary of the biological implications.

3. In the Discussion section we suggest that the INSIG-SREBP-2 pathway is likely to proceed via a similar mechanism in both mouse genotypes.